# Effects of Velocity-Based Training on Strength and Power in Elite Athletes—A Systematic Review

**DOI:** 10.3390/ijerph18105257

**Published:** 2021-05-14

**Authors:** Michał Włodarczyk, Przemysław Adamus, Jacek Zieliński, Adam Kantanista

**Affiliations:** 1Department of Athletics, Strength and Conditioning, Poznan University of Physical Education, 61-871 Poznań, Poland; jzielinski@awf.poznan.pl; 2Independent Researcher, 63-000 Wielkopolska, Poland; adamusp83@gmail.com; 3Department of Physical Education and Lifelong Sports, Poznan University of Physical Education, 61-871 Poznań, Poland; kantanista@awf.poznan.pl

**Keywords:** speed, resistance, velocity loss, velocity zones, feedback

## Abstract

Due to drawbacks of the percentage-based approach, velocity-based training was proposed as a method to better and more accurately prescribe training loads to increase general and specific performance. The purpose of this study was to perform a systematic review of the studies that show effects of velocity-based resistance training on strength and power performance in elite athletes. Electronic searches of computerized databases were performed according to a protocol that was agreed by all co-authors. Four databases—SportDiscus with Full Text and MEDLINE via EBSCO, SCOPUS, and Web of Science—were searched. Seven studies were found which researched the effects of velocity-based resistance training on athletes after a given training period. The analyzed studies suggest that applying velocity losses of 10–20% can help induce neuromuscular adaptations and reduce neuromuscular fatigue. Using velocity zones as part of a separate or combined (e.g., plyometric) training program can elicit adaptations in body composition and performance parameters. Moreover, velocity zones can be programmed using a periodized or non-periodized fixed velocity zones protocol. Lastly, obtaining instantaneous feedback during training is a more effective tool for increasing performance in sport-specific parameters, and should be used by sport practitioners to help keep athletes accountable for their performance.

## 1. Introduction

Increasing athletic performance requires constant development of strength and power parameters [1]. To develop strength and power, different training methods (e.g., weightlifting, plyometric training, eccentric training, and ballistic training) can be implemented [1]. A relatively new method for strength and power development is velocity-based training (VBT). In VBT, movement velocity of an exercise is tracked using a linear position transducer (LPT) [2,3,4] and used to measure exercise intensity. Using VBT real-time feedback is obtained on a repetition-to-repetition basis. Thus, coaches and athletes may establish the speed of the movement at which an exercise is performed [5].

VBT is a relatively novel method and can be an alternative to the “percent-based” approach where exercise intensity or load is prescribed as a relative load (percentage of one-repetition maximum, % of 1RM) [5]. Efficacy of VBT is still being verified and is continuously under debate [6,7].

VBT is superior to percentage-based training (PBT) especially in elite athletes since PBT has many limitations: firstly, in order to prescribe training loads a 1RM test must be conducted which may lead to injury when performed incorrectly [5]. It is also time consuming and impractical in large group settings, as well as the obtained values contain a large margin of error [3]. Secondly, changes in 1RM can take place even after a few training sessions and can fluctuate on a day-to-day basis depending on daily readiness caused by normal biological variability, fatigue associated with training or life-style factors, like sleep, stress, and nutrition [3,5]. Lastly, changes in 1RM may not reflect changes within the force-velocity continuum which may warrant a need to monitor changes in individual load-velocity profiles of athletes together with 1RM changes [5]. Therefore, VBT was proposed as a method to more accurately prescribe training loads to increase general and specific performance. Since elite athletes undergo training at very high intensities and volumes, VBT can be a more precise and accurate method to prescribe training load.

VBT can be used in many ways; Jovanovic and Flanagan [5] proposed many applications for sport practitioners. One important application is to apply velocity loss thresholds and to create velocity-exertion profiles to monitor fatigue and exertion [3,5,8]. Velocity loss is a parameter which reflects the relationship between metabolic fatigue present in a working set. The closer the set is taken to failure or in another words the more reps that are performed in respect to the maximum amount of reps that can be performed the greater the velocity loss [8]. Applying velocity loss thresholds or “cut-off velocities” can limit the amount of fatigue induced and overall control training volume depending on the training goal [5]. Another way to use VBT is to prescribe a certain target velocity to obtain during each repetition, or to aim for a velocity range or zone. This especially can be used when trying to maximize power production in a certain exercise [9,10]. Since there is a relationship between movement velocity and relative intensity (as a percentage of 1RM) maintaining an appropriate velocity zone can insure an athlete is working at a stable working intensity (compared to PBT, where percentages can fluctuate on a daily basis) [3]. Lastly, velocity can be used as a feedback tool to increase performance through increased competitiveness and motivation [11,12]. By receiving immediate feedback from a performed set the sport practitioner can also modify the load used to match the athlete’s daily readiness. Therefore, VBT was proposed as a method to more accurately prescribe training loads to increase general and specific performance.

Guerriero et al. [7] analyzed the existing literature on the effects velocity based training and the common methods utilized by elite athletes. Authors concluded that resistance training and velocity monitoring can be effective in enhancing sport specific performance together with endurance and power training in elite athletes. However, they only analyzed studies including a resistance training protocol of more than four weeks and a literature search was conducted on different electronic databases and up to July 2018. In the present study we focused on how VBT methods are implemented in elite training environments and also included studies published until 1 March 2020. We divided protocols used into either the velocity loss method, velocity zone method, or the velocity feedback method. This study can help present current research on the effects of VBT on strength and power in athletes and to further recommend guidelines for coaches on how to apply this novel training practice.

The purpose of the present paper is to perform a systematic review of the studies that show effects of velocity-based resistance training on strength and power performance in elite athletes.

## 2. Materials and Methods

### 2.1. Search Strategy

Electronic searches of computerized databases were performed according to a protocol for this review that was agreed by all co-authors. Four databases—SportDiscus with Full Text and MEDLINE via EBSCO, SCOPUS, and Web of Science—were searched. A literature search for papers was carried out using keyword combinations: (1) Velocity-Based Resistance Training Or Velocity-Based Strength Training; (2) Athletes Or Players or Competitors; (3) Effects Or Impact Or Consequences Or Influence Or Outcomes. Boolean operators (AND, OR) were used to concatenate the search terms. A secondary search was performed by screening the reference lists of the included studies and relevant review articles. The study selection process is presented in Figure 1.

Papers were reviewed if they met the following criteria: original research published in English, published until 1 March 2020 as a full-text manuscript, papers in which effects of velocity-based resistance training on strength and power in athletes were shown, pre- and post-assessments were performed; participants included in the studies were 15 years old or older and competed at least at the national level. “Grey” literature or unpublished studies were not analyzed.

### 2.2. Data Extraction

The search strategy was run by AK with expertise in systematic reviews. Articles were extracted and imported into EndNote software. Duplicate articles were removed using Endnote. Titles and abstracts of potentially relevant articles were analyzed by two reviewers (AK and PA). Full text of papers were then obtained for those meeting initial screening. Then the full texts of copies of the papers were analyzed independently. If an article was included by one reviewer, and not the other, the article was obtained for further review by a third reviewer (MW). From the initially accepted list of papers information concerning Author, Number of subjects, sex, age, Training experience, sport-specific background, Frequency, Duration, Purpose of the study, Use of velocity in the training protocol, Training effects were extracted and recorded. During meetings of all co-authors, extracted information was discussed and papers which presented the same research results were excluded. Afterwards, the final list of papers were accepted. [13]

### 2.3. Risk of Bias

The Jaded scale, a five-point quality scale was used to independently assess the quality of the trials and to designate a score between zero (very poor) and five (rigorous) [14]. Questions regarding randomization, double blinding and description of withdrawals and dropouts were answered and accordingly marked (0–2 points for questions one and two each, and 0–1 point for question three). This was done for all studies in the review giving each study a score of 0–5. A minimum cut-off inclusion score was not applied in order to maximize the amount of studies included in the review.

## 3. Results

### 3.1. General Characteristics of the Studies

Finally, we found seven studies which researched the effects of velocity-based resistance training on athletes after a given training period (Table 1). Overall, in these studies 166 participants were enrolled. Only in one study, the research group was represented by females. Most studies were carried out on groups of soccer players (four studies). Flat-water kayak paddlers, rugby, and volleyball players were also included in these studies. It is important to note that this review was not limited to studies under 4 weeks; however, no such studies with national level athletes were found which could be included.

### 3.2. Velocity Loss Method

In two studies [15,18] the velocity loss method was used, where the set of the given exercise is stopped after movement velocity falls below a certain threshold. Garcia-Pallares et al. [15] analyzed changes in selected cardiovascular and neuromuscular variables during a 12-week training cycle. Using this periodized cycle, eleven world-class level paddlers underwent a battery of tests four times (T0, T1, T2, and T3). This study included tests such as anthropometric measurements, kayak-ergometer and resistance exercises tests. Velocity based resistance training was applied in the bench press and prone bench pull exercises. These two exercises were chosen because they are typical resistance training exercises for this group of athletes. Firstly, 1RM was determined for each subject, then mean concentric velocity with 45% of the previously determined 1RM load was assessed for both exercises. Subjects performed three different training sessions per week: a hypertrophy, maximal strength and maximal power training session. In maximal power training sessions, each set was terminated when mean velocity decreased more than 10% of the fastest repetition’s mean concentric velocity. For the bench press, from T0 to T1 1RM improved significantly (9.7%), while mean concentric velocity with 45% of the previously determined 1RM load (V45%) remained unchanged. Between T1 and T2, no significant changes were observed in 1RM, but V45% were improved (5.3%). From T2 to T3, 1RM values significantly decreased (4.6%), while V45% significantly improved by 11.0%. When comparing T0 to T3, significant improvements were found in 1RM (4.2%) and V45% (14.4%). For the prone bench pull, between T0 and T1, 1RM improved significantly (7.7%) and V45% remained unchanged. From T1 to T2 only V45% improved (4.6%). From T2 to T3 values significantly decreased (4.5%) and V45% significantly improvement by 7.1%. Between T0 and T3, significant improvements were found in 1RM (5.3%) and in V45% (10.0%).

Pareja-Blanco et al. [18] showed differences on performance tests after 6 weeks training, three times per week in highly trained soccer players. In this training program, the same relative loading was used, but repetition volume was different. Two groups: the 15% repetition velocity loss group (VL15) and the 30% repetition velocity loss group (VL30) were created. Velocity loss during the set was independent variable for this study (15%—VL15 and 30%—VL30 of velocity loss). Both groups performed 18 resistance training sessions where they focused on velocity-based squat training program. This program based on target mean propulsive velocity (MPV), which was given for every session and percentage velocity loss from assumption target. Subjects performed pre- and post-tests, including: 1RM estimate from isoinertial squat loading test, change in average MPV (AMPV), countermovement jump (CMJ), 30 m sprint time (T30) and yo-yo intermittent recovery test 1 (YYIRT). VL15 group showed a better effects on 1RM strength and average mean propulsive velocity than VL30 group. V15 group showed significantly greater gains in CMJ, whereas in VL30 group noticed a possibly negative effect on CMJ performance. The effects on T30 performance were unclear/unlikely for VL15 and VL30, but YYIRT showed most likely/likely positive effects in both groups.

### 3.3. Velocity Zones Method

Other four studies [16,17,19,20] showed the velocity zones method, when the velocity of repetition was known and subjects had to do repetitions in a given range of velocity.

Gonzalez-Badillo et al. [16] examined 44 young soccer players to analyze the effect on physical performance, after a 26-week velocity-based resistance training period. All three teams, pre and post training intervention, performed the following tests: 20 m running sprint (T20), CMJ, a progressive isoinertial loading squat test to determine load at 1 m/s (V1LOAD) and maximal aerobic speed (MAS). A strength training program using full squats and squat jumps with load was applied. Furthermore, assistance exercises, for example box jumps, or sled towing were also performed. Each player was instructed to perform the full squat with the V1LOAD and a load which the athletes was able to jump 20 cm in the countermovement jump (CMJ_L_). Results showed, that a velocity based training program improved performance variables. V1LOAD significantly improved more in U16 group than in other groups. U16 obtained significantly higher increases in CMJ height compared to players from U21. U18 also shows greater increases in CMJ than U21. In the T20 test, effects were unclear. Moreover, in MAS, U16 group showed greater increases than U21. It should be noted that correlations were found between changes in CMJ with changes in T20 and V1LOAD.

Lopez-Segovia et al. [17] assessed the effect of velocity-based resistance training on strength, acceleration capacity and aerobic power. Two under-19 Spanish soccer teams completed a 16-week training program. Subjects had two evaluations, before and after the training period, where CMJ and CMJ_20_, Smith machine bar movement speed (FS_L_), acceleration capacity at various split times and MAS were tested. Loads used by each players were individually determined according to the results of the initial test. The progression was performed with the objective of players working with a 1 m/s bar speed in the FS, corresponding to around 55% of 1RM. Overall, team A improved in the CMJ (5%), FS_20_ and highly significantly in FS_30_ and FS_40_, as well as an improved MAS. Team A improved significantly in most timed splits. 

Rauch et al. [19] were measured the effect on strength and power adaptations, using two different training regiments: progressive velocity based training (PVBT) and optimal training load (OTL). The subjects were female collage volleyball players, which trained three times per week for seven weeks (weight training on Monday, Wednesday, and Friday). All subjects completed six familiarization sessions, two for each exercises: back squat (BS), bench press (BP) and deadlift (DL). Additionally, lean body mass and fat mass were evaluated via dual-energy X-ray absorptiometry (DXA). Players also underwent baseline testing on jump height (CMJ and squat jump (SJ)), agility (*t*-test), 1RM, and submaximal peak power (PP) assessment on BS, BP, and DL. Both groups were re-tested during week 8. The strength training program was different for both groups. PVBT group performed a 4-week strength block (0.55–0.70 m/s) followed by a 3-week power block (0.85–1.0 m/s). OTL trained at 0.85 m/s, 0.85 m/s, and 0.9 m/s on the BS, BP and DL respectively for weeks 1–7. In addition, four accessory exercises was included for each training day, performed in a circuit. Results showed main time effect for BS 1RM (PVBT: 19.6%; OTL: 18.3%), BP 1RM (PVBT: 8.5%; OTL: 10.2%), DL 1RM (PVBT: 10.9%; OTL:22.9%), BS PP (PVBT: 18.3%; OTL: 19.8%), BP PP (PVBT: 14.5%; OTL: 27.9%), and DL PP (PVBT: 15.7%, OTL: 20.1%). Additionally, both groups increased in lean body mass, and decreased in fat mass. No significant changes were observed in CMJ or SJ. In agility test both groups decreased trial times.

Rodriguez-Rosell et al. [20] analyzed the effects of light-load maximal lifting velocity weight training (WT) combined with plyometric training (PT), and compared this training against WT alone. 30 adult soccer players were assigned into three groups: strength training group only performing the full squat (FS) exercise (FSG), combined group performing the full squat with jump and sprint exercises (COM) or control group (CG) which only performed soccer training. FSG and COM trained twice per week for 6 weeks using free-weight FS (both groups), as well as jumps, sprints and changes od directions (COD) (only COM). All groups performed four field soccer training sessions and a friendly match every week. All subjects were evaluated before (pre) and after the 6-week strength training routine with the following tests: (1) 20 m all-out sprints (T_10_, T_20_, T_10-20_); (2) CMJ, (3) a progressive isoinertial loading full squat (FS) test. Four pre-testing familiarization session were carried out for subjects to properly learn to perform the FS and CMJ exercises. Strength training was performed before field training. Loads for the FS were determined according to the movement velocity obtained during the isoinertial squat loading test. The target velocity was set at 1.20 m/s and decrease to 1.00 m/s throughout the 6-week training period. Subjects were instructed to perform the required movements with maximal intended velocity. Both training groups showed a significantly greater percentage of change than CG in 1RM_est_ (*p* < 0.05–0.01), mean propulsive velocity at 30 kg (MPV30) (*p* < 0.05), mean propulsive velocity at 40 kg (MPV40) (*p* < 0.01), T_20_ (*p* < 0.05–0.001) and CMJ (*p* < 0.05–0.001). No significant differences were found between FSG and COM for any variable. The authors concluded that both experimental groups had significant increases in all variables, compared to the CG which had no significant changes.

### 3.4. Velocity and Feedback

Only one study [12] showed the important role of feedback to perform more effective repetitions and obtain greater results in performance tests. Randell et al. [12] investigated the importance of feedback in 6-weeks velocity-based resistance training. Authors tried to show differences in effects between the same training program, but with- and without feedback. Subjects performed concentric squat jumps, 3 sets of 3 repetitions. The feedback group received visual real-time feedback on peak velocity of the jump squat using a linear position transducer. Sport-specific performance tests results pre- and post-training program showed a greater benefits for feedback group, but the effects were small. Only in 30 m sprint time and horizontal jump were statistically significant differences between group.

## 4. Discussion

The aim of this study was to perform a systematic review of the studies that show effects of VBT on strength and power performance in athletes. Based on the results of this review it can be noted that VBT can be used in many ways: the velocity loss method, velocity zones method, and velocity and feedback method to increase general and specific performance. 

### 4.1. Velocity Loss Method

A relationship between velocity and proximity to failure when programming resistance training loads was established [5,8,21,22]. The closer a set is taken to failure, the larger the velocity loss [8]. The velocity loss method is based on the premise that if a velocity cut-off point is set, this will limit the amount of neuromuscular fatigue that will take place during a set. This in turn will help maintain a higher quality of neuromuscular work (strength, power, speed) performed during a training session and ultimately increase specific performance (sprint, jump, change of direction performance, etc.). Velocity losses/thresholds of 10% [15], 15%, and 30% [18] were used in the reviewed studies. 

In all reviewed studies, significant changes in performance parameters (e.g., 1RM, max speed, CMJ) were noted when utilizing a velocity loss and in studies comparing two different velocity losses [18] the smaller velocity losses elicited greater neuromuscular adaptations compared to the larger velocity losses (15% vs. 30%). It is clear that smaller velocity losses allow greater quality of repetitions to be performed with consequently higher velocities achieved throughout the exercise session. Parejo-Blanco et al. [18] also studied performance gains (1RM, CMJ, 20 m sprint running, full load-velocity squat profile) as well as cross sectional area changes and muscle fiber analysis when utilizing velocity losses of 20% or 40% in young males. They concluded that the 40% velocity loss group obtained greater increases in cross sectional area, and a reduction in the percentage of myosin heavy chain IIX, while the 20% velocity loss group obtained greater increases in CMJ performance and maintained the percentage of myosin heavy chain IIX muscle fibers. Both groups had similar increases in squat strength, however the 20% velocity loss group performed 40% less repetitions. Moreover, Padulo et al. [23] compared the effects of fixed pushing speed (FPS), when the velocity loss was 20% and self-selected pushing speed (SPS), when subject tried to do repetitions to exhaustion, in the bench press exercise. Twenty resistance-trained subjects, with training experience participated in this study. This study showed, that FPS training had greater benefits in increasing muscle strength than SPS training. Furthermore, FPS training increased concentric phase maximal speed, better than SPS training. Therefore, greater velocity losses will create different functional and structural neuromuscular adaptations compared to smaller velocity losses.

Additionally, Sanchez-Medina and Gonzales-Badillo [8] studied velocity losses, CMJ height decreases and the metabolic response (lactate and ammonia concentration) to various resistance exercise protocols with various levels of exertion (proximity to failure). They observed that the greater the number of repetitions performed and even more so the greater number of repetitions closer to failure (greater exertion) the greater the velocity loss, the greater the CMJ decrease, and the greater the lactate and ammonia concentration accumulation. The also noted a correlation between velocity loss and lactate (linear relationship) and ammonia (curvilinear relationship) concentration meaning the greater the velocity loss, the greater the metabolic response. This implies that by utilizing velocity loss threshold/cut-off points, the sport practitioner can also limit metabolic stress of a given exercise session. The present review shows that velocity losses of 10%, 15%, and 20% all induce neuromuscular adaptations and athletes focusing on increasing neuromuscular performance should apply a range between 10 and 20% velocity loss to maintain optimal quality of work.

### 4.2. Velocity Zones Method

Many previous studies have shown a relationship between load and velocity which can be presented as the force-velocity profile [3,5,22,24,25,26]. The load-velocity or force-velocity profile demonstrates that with increasing load, the velocity should decrease and vice-versa. This means that in a given exercise, as the load increases, as long as intent to perform the exercise “as fast as possible” during the concentric phase is maximal, velocity should decrease. Each exercise also has a minimal velocity threshold (MVT), which is the mean concentric velocity produced on the last repetition of a set to failure performed with maximal lifting effort [5]. MVT has also been shown to remain stable even when strength increases in a given exercise [3]. By obtaining MVT of an exercise and creating a load-velocity profile for an athlete in one or a few key exercises, this allows the sport practitioner to track progress over time, across the full spectrum of force-velocity qualities. 

These qualities can help to better compare two athletes and is especially important when the sport practitioner is interested in velocity specific adaptations to training and not exclusively on maximal strength [5]. These velocity specific adaptations can be divided into velocity zones, which can dictate the way training is programmed. Loturco et al. [10] studied whether there are mean propulsive velocities capable of maximizing mean propulsive power in four exercises: half squat, jump squat, bench press and bench throw. They concluded that by using a linear position transducer (LPT) and the four mean propulsive velocities provided (0.93 m/s, 1.02 m/s, 1.40 m/s, and 1.67 m/s for the half squat, jump squat, bench press, and bench throw, respectively) sports practitioners can adjust the training loads on a daily basis when performing the mentioned exercises. This has value since it is possible to prescribe appropriate training loads depending on goal (e.g., maximal power output) and monitor progress on a daily basis. In this review, all four studies utilized a set velocity zone (velocity goal), which had to be met each training session. Athletes were required to adjust load to the prescribed velocity.

Gonzales-Badillo et al. [16], and Lopez-Segovia et al. [17] both used velocity zones around 1.0 m/s (full squat exercise) and maintained this velocity zone throughout the whole training program. Rauch et al. [19] divided subjects into two groups; one with a progressive velocity based training protocol of increasing velocity (4 week strength block with 0.55–0.70 m/s velocity zone, and later a 3 week power block with 0.85–1.0 m/s velocity zone) and another training protocol with constant optimal training load (0.85–0.9 m/s velocity zone). Lastly, Rodriguez-Rosell et al. [20] used a velocity zone between 1.20 and 1.00 m/s where the athletes began at 1.20 m/s and every three sessions the prescribed velocity was decreased (1.12 m/s, 1.06 m/s, and 1.00 m/s). Gonzalez-Badillo et al. [16] and Lopez-Segovia et al. [17] both concluded that a resistance training protocol with a set velocity zone was effective in increasing performance parameters while decreasing the amount of reps performed. Rauch et al. [19] and Rodriguez-Rosell et al. [20] both had groups with a periodized progression of velocity zones. Rauch et al. [19] concluded that both periodized and fixed velocity zones protocol were effective in improving body composition and performance parameters in volleyball players. These findings question the need for a preliminary strength block to induce body composition and performance adaptations in such a short time frame (6–8 weeks). It would interesting to see if periodization of velocity zones would influence greater adaptations if a larger time frame was used (e.g., >12 weeks). Rodriguez-Rosell et al. [27] also used a periodized approach by decreasing the velocity zone target every three sessions; however, they only tested the results of an only strength training (FSG), vs. combined (COM) (strength training, jumps, sprints, and changes of direction) training program vs. a control group who only performed sport-specific (soccer) training (no comparison between training protocols). 

According to this review, using velocity zones as part of a separate or combined (with plyometric) training program can elicit adaptations in body composition and performance parameters. Velocity zones can be programmed using a periodized approach (e.g., strength blocks with velocities of 0.55–0.7 m/s, and power blocks with velocities of 0.85 m/s) but will depend on the exercise chosen (velocities producing maximal power will differ for exercises such as the half squat, jump squat, bench press, and bench throw). When dealing with short [6,7,8] training periods meant to increase multiple physical attributes, a non-periodized fixed velocity zone protocol is just as effective.

### 4.3. Velocity and Feedback

Performance feedback is an essential element in the training process, and usually is manifested in the form of a coach correcting technique, measuring the height of a jump, time of a sprint, etc. With VBT resistance training, obtaining feedback in the form of velocity of the repetition performed (parameters such as average or peak velocity) can also serve to improve performance. Only one study [12] in this review analyzed the effect of instantaneous performance feedback (peak velocity of 40 kg concentric squat jumps) vs. a non-feedback group (40 kg concentric squat jumps) on sport specific performance tests (vertical and horizontal jumps, as well as 10/20/30 m timed sprints). The researchers concluded that the instantaneous feedback group obtained greater improvements in sport-specific performance tests compared to the non-feedback group. Although there is still little research studying VBT feedback on performance in athletes, the practical application of doing so is invaluable. 

When training athletes, increasing the amount of high quality reps, translates into better performance in the long-term. Velocity feedback during training not only motivates athletes to achieve better results but keeps them accountable for their performance. Velocity feedback also can be used as a coaching tool to help athletes understand corrections that may need to take place in their technique when performing an exercise (e.g., higher velocity achieved when a power exercise is performed properly). Velocity feedback also helps to monitor changes in strength similarly to the velocity zones method (trying to keep within a desired range) where progress can be observed if at a similar load a higher velocity is achieved. Another important aspect of using velocity as feedback is to verify if an athlete is compliant and adhering to a training program that was created for them. If the coach or sport practitioner knows a certain athlete’s performance capabilities (through objective testing), then they can hold the athlete accountable to perform the exercise with the required velocity. A few limitations of VBT include 1) the need to purchase rather expensive equipment (LPTs) which may discourage coaches with low budgets, and 2) athletes need to be proficient in strength training exercise technique in order for the obtained results to have valid meaning. Overall, according to this review, obtaining instantaneous feedback during training is an effective tool for increasing performance in sport-specific parameters, and should be used by sport practitioners to help motivate athletes for better adaptations, and keep them accountable for their performance.

### 4.4. Study Limitations

There are a few minor limitations that must be taken into account. Firstly, some of the reviewed studies did not use velocity based resistance training as the main training intervention but rather as part of a combined program. Therefore, it is difficult to assess whether velocity based resistance training as an isolated training intervention would produce the same results. However, this adds practical value since very rarely in sports physical training (strength and conditioning) are isolated training interventions used. Next, due to the small number of studies using velocity based resistance training in strength and power training and since most of these studies measured velocity in various exercises and protocols it is difficult to assess the effectiveness of a training intervention using velocity based resistance training. More studies utilizing the same protocols (ex/same velocity losses, velocity targets/zones) are needed to verify effectiveness of velocity based resistance training methods in athletes. Lastly, most studies scored very low on the Jadad scale (Table 1), which can indicate a low quality of trials. Performing randomization and double blind protocols in sport science studies utilizing a training intervention is very difficult since the content of the training sessions is known by the athletes. 

## 5. Conclusions

According to the analyzed studies, a few important recommendations regarding the implementation of a VBT training program can be made: (1) applying velocity losses of 10–20% can help induce neuromuscular adaptations and reduce neuromuscular fatigue associated with maintaining better/higher quality of work; (2) using velocity zones as part of a separate or combined (e.g., plyometric) training program can elicit adaptations in body composition and performance parameters; (3) velocity zones can be programmed using a periodized (strength blocks, and power blocks) or non-periodized fixed velocity zones protocol (non-periodized protocol is just as effective as periodized when limited to short training periods meant to increase multiple physical traits); and (4) obtaining instantaneous feedback during training is a more effective tool for increasing performance in sport-specific parameters, and should be used by sport practitioners to help keep athletes accountable for their performance. 

## Figures and Tables

**Figure 1 ijerph-18-05257-f001:**
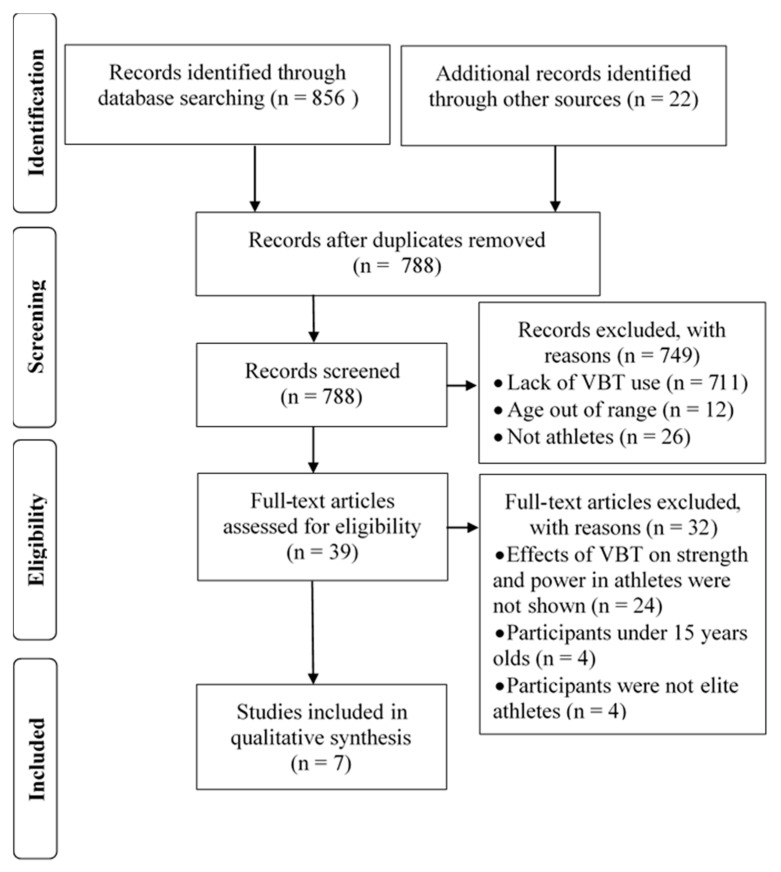
Flow diagram for the identification, screening, eligibility, and inclusion of studies.

**Table 1 ijerph-18-05257-t001:** Characteristics of studies included in review.

Author	Number of Subjects, Sex, Age (Years)	Training Experience, Sport-Specific Background	Frequency (Sessions/Week) Duration (week)	Purpose of the Study	Use of Velocity in the Training Protocol	Training Effects	Jadad Scale (Points)
Garcia-Pallares et al. 2009 [15]	11M; 26.2 ± 2.8	World-class, flat-water kayak paddlers; 12.4 ± 2.1 (years)	3 per week; 12 weeks	Examine the effects brought about by a 12-week periodized program of combined strength and endurance training on selected neuromuscular and cardiovascular parameters	In maximal power training sessions (P3), in BP and PBP exercises, each set was terminated when mean velocity decreased by more than 10% of the best (fastest) repetition’s mean concentric velocity	Significant improvements: 1RM in BP (4.2%) and PBP (5.3%), V45% in BP (14.4%) and PBP (10.0%) were observed from T0 to T3	0
Gonzalez-Badillo et al. 2015 [16]	U16 = 17M; 14.9 ± 0.3 U18 = 16M; 17.8 ± 0.4 U21 = 11M; 19.2 ± 1.2	Soccer players	U16 and U18—2 RT sessions per week; 26 weeks U21—only typical soccer training	Analyze the effect of velocity-based resistance training with moderate loads and few repetitions per set combined with jumps and sprints on physical performance in young soccer players of different ages	Isoinertial progressive loading test were performed to assess V1LOAD for every player; squats load in training program based on V1LOAD	U16 > U18 & U21 in V1LOAD U16 & U18 > U21 in CMJ height U16 > U21 on MAS	1
Lopez-Segovia et al. 2010 [17]	Team A: 19M; 18.43 ± 0.6 Team B: 18M; 18.08 ± 0.8	Under-19 Spanish first division soccer players	1–2 per week; 16 weeks	Assess the effect of the training on aerobic power, strength, and acceleration capacity	The players always work with a load that they were able to lift in a FS at approximately 0.8–1.0 m/s, velocity depends of training week	Team A: ↑ MAS, ↑ CMJ_20,_ ↑ FS_20-30-40,_ ↓ acceleration capacity in all the splits Team B: ↓ MAS, ↑ CMJ_20,_ ↑ FS_50-60,_ ↑ T_20-30_	0
Pareja-Blanco et al. 2016 [18]	16M; 23.8 ± 3.5	Highly trained soccer players	3 per week; 6 weeks	Analyze the effects of two RT programs that used the same relative loading but different repetition volume using the velocity loss during the set: 15% (VL15) vs. 30% (VL30)	Two groups: VL15 & VL30 had identical training session (squat RT program), with the same relative loading magnitude (%1RM), but differed in the max percent velocity loss reached in each exercise set (15% vs. 30%)	CMJ height: VL15 > VL30 VL15: likely/possibly positive effect on 1RM, AMPV, CMJ VL30: possibly/unclear positive effects on 1RM, AMPV; possibly negative effects on CMJ VL15 and VL30: unclear/unlikely effects on T30; most likely/likely positive effects on YYIRT VL15 is effective to induce improvements in neuromuscular performance	1
Randell et al. 2011 [12]	Feedback group: 7M; 25.7 ± 3.6 Non-feedback group: 6M; 24.2 ± 2.5	Professional rugby players; Feedback group: 3.7 ± 1.0 (years) Non-feedback group: 3.2 ± 1.2 (years)	3 per week (squat jumps: 2 per week); 6 weeks	Investigate the effect of instantaneous performance feedback (peak velocity) on sport-specific performance tests	Concentric squat jumps: 3 sets of 3 repetition with feedback and without feedback + typical preseason conditioning program	Small effects, expect for the 30 m sprint performance, which was moderate; feedback group increased the results on sport-specific performance tests more than non-feedback group	1
Rauch et al. 2018 [19]	15F; 19.3 ± 1.4	Collegiate volleyball players	3 per week; 7 weeks	Investigate the effects of two different VBT regimens on muscular adaptation (PVBT and OTL)	PVBT group: 4-week strength block (0.55–0.70 m/s); 3-week power block (0.85–1.0 m/s); OTL group: 7 weeks of BS (0.85 m/s), BP (0.85 m/s), DL (0.9 m/s); both groups performed accessory exercises in a circuit	BS 1RM: PVBT: ↑ 19.6%, OTL: ↑ 18.3% BP 1RM: PVBT: ↑ 8.5%, OTL: ↑ 10.2% DL 1RM: PVBT: ↑ 10.9, OTL: ↑ 22.9% BS PP: PVBT: ↑ 18.3%, OTL: ↑ 20.1% BP PP: PVBT: ↑ 14.5%, OTL: ↑ 27.9%, DL PP: PVBT:↑ 15.7%, OTL: ↑ 20.1%	1
Rodriguez-Rosell et al. 2017 [20]	30M; 24.5 ± 3.4	Spanish third division semiprofessional soccer players	2 per week; 6 weeks	Compare the effects of combined light-load maximal lifting velocity weight training and plyometric training with weight training alone on strength, jump and sprint performance	FSG (*n* = 10): FS only, load progressively increased from ~1.20 m/s (~45%1RM) to ~1.00 m/s (~58%1RM) COM (*n* = 10): FS combined with jumps, sprints and changes of direction CG (*n* = 10)	1RM: ↑ (17.4–13.4%); CMJ: ↑ (7.1–5.2%); Sprint time: ↑ (3.6–0.7%); Force-velocity relationships: ↑ (16.9–6.1%); no significant differences between FSG and COM	1

Note: P3, third training phase; VO_2max_, maximal oxygen uptake; VT2, ventilatory threshold; PS_max_ paddling speed at VO_2max_, paddling speed at VT2; 1RM, one repetition maximum; BP, bench press; PBP, prone bench pull; T0, first date of tests during training cycle; T3, last date of tests during training cycle; U16, under-16 team; U18, under-18 team; U21, under-21 team; RT, resistance training; V1LOAD, the load that elicited 1.00 m/s velocity in the full squat exercise; CMJ, countermovement jump; MAS, maximal aerobic speed; FS, full squat; CMJ_20_, countermovement jump with 20 kg; FS_20–30–40_, full squat with load: 20, 30, 40 kg; FS_50–60_, full squat with load: 50, 60 kg; T_20–30_ acceleration capacity between 20 and 30 m; VL15, group that trained with a mean velocity loss of 15% in each set; VL30, group that trained with a mean velocity loss of 30% in each set; AMPV, average mean propulsive velocity attained against absolute loads common to Pre- and Post-tests in the squat progressive loading test; T30, 30-m sprint; YYIRT, Yo-yo intermittent recovery test level 1; VBT, velocity-based training; PVBT, progressive velocity-based training; OTL, optimum training load; BS, back squat; DL, deadlift; PP, peak power output; FSG, full squat group; COM, combined group; CG, control group.

## Data Availability

Not applicable.

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
