# Peer review of "Effects of Velocity-Based Training on Strength and Power in Elite Athletes—A Systematic Review"

_ijerph, 2021, doi:10.3390/ijerph18105257_

Round 1

Reviewer 1 Report

Comments and suggestions for authors

First of all, I would like to thank the authors for their work. The review presented is done with professional respect for the authors of the manuscript, with the aim of improving it.

INTRODUCTION

L28-29: Provide a reference

L28-35: Each sentence speaks of a new subject... the text must be cohesive.

L41-41: Provide a reference

L45: Why focus in novice athletes?

L63-65: Remove sentence: in a theoretical framework it cannot make an opinion of the authors.

L69: “In the present study we focused on how VBT methods are implemented in elite training” however, L72: “The purpose of the present paper is to perform a systematic review of the studies that 75 show effects of velocity-based resistance training on strength and power performance in 76 athletes”. Where is the focus of the review, on athletes or elite athletes?

NOTE: It is very difficult to follow an adequate line explaining the different arguments that the authors have used to conceptualize the work, very difficult. There are constant unrelated changes. Recommendation: Rewrite the introduction. It is not necessary to write a long introduction, it is better to write a short and clear introduction, which helps to contextualize the objective of the study.

MATERIAL AND METHODS;

L83: Athletes, players or competitors? It will depend on the ultimate objective that the authors decide to set.

Figure 1: Specify and detail records excluded and full text articles excluded.

RESULTS

L145: “For the purpose of our study the most important conclusion was the results of using velocity based resistance training in the bench press and prone bench pull exercises”… Are you copying sentences directly from the articles reviewed?

Section 3.3 Complete rewrite, has a lot of text copied from: https://journals.lww.com/nsca-jscr/Fulltext/2010/10000/Effect_of_4_Months_of_Training_on_Aerobic_Power,.18.aspx

Reviewer 2 Report

Point-by-point

The present manuscript provides a systematic review on the effects of resistance training using a Velocity Based Resistance Training Approach (VBRT) as methodology to control the different resistance training variables on strength and power performances in athletes from different sport disciplines. This is an interesting topic as it provides precision in the control and quantification of the training load as well as the possible effects of different training programs.

Specific article comments:

  1. Comment (Title): The authors use the noun athletes in the title. This can be understood as "one who participates in physical exercise or sports, especially in competitive events", regardless of the level of these. However, the population in which the final selection of the articles seems to be cantered seems to be more related to athletes of a medium-high level of performance. Authors might consider adjusting this question in the title (e.g. elite athletes).
  2. Comment (Abstract): Overall, the abstract is clear.
  3. Comment (Introduction).

Well written with a rationale justification for the revision. However, in my opinion, it is necessary to develop on what is based the methodology by which the review is being carried out (i.e. VBRT). It is important for the reader to know which are the bases on which this methodology is based. In summary, it could be appropriate to refer to the relationship between movement velocity and the relative intensity observed in different exercises, to justify its use as an indicator of relative intensity. In the same way, mention could be made of the relationship between the velocity loss that takes place in the series and the percentage of repetitions performed with respect to the maximum number of repetitions that can be performed to explain the use of the velocity loss as a variable to control the training volume. Therefore, I consider it appropriate to mention the methodological bases of this approach.

  1. Comment (Methods): Overall, the method is clear.
  2. Comment (Results): Overall, the results are clear.
  3. Comment (Discussion): Overall, the results are clear.
  4. Figure and table: Overall, the table is clear.

Round 2

Reviewer 1 Report

Dear authors;

I appreciate the changes made, however the manuscript continues to have a problem with section 3.3.

In the following link (https://drive.google.com/file/d/1xGhuv9LVzJeio01DehMDVcsQDwiB8pj7/view?usp=sharing), you can see that the text marked in red belongs to the article (https://journals.lww.com/nsca-jscr/Fulltext/2010/10000/Effect_of_4_Months_of_Training_on_Aerobic_Power,.18.aspx).

You should rewrite this section

Best regards
